# Characteristics and Applications of Canine In Vitro Models of Bladder Cancer in Veterinary Medicine: An Up-to-Date Mini Review

**DOI:** 10.3390/ani12040516

**Published:** 2022-02-19

**Authors:** Łukasz Nowak, Wojciech Krajewski, Bartosz Małkiewicz, Tomasz Szydełko, Aleksandra Pawlak

**Affiliations:** 1University Center of Excellence in Urology, Department of Minimally Invasive and Robotic Urology, Wroclaw Medical University, 50-556 Wroclaw, Poland; wk@softstar.pl (W.K.); bartosz.malkiewicz@umed.wroc.pl (B.M.); tomasz.szydelko1@gmail.com (T.S.); 2Department of Pharmacology and Toxicology, Faculty of Veterinary Medicine, Wroclaw University of Environmental and Life Sciences, 50-375 Wroclaw, Poland

**Keywords:** canine, bladder, urothelial cancer, in vitro model

## Abstract

**Simple Summary:**

Bladder cancer (BC) in dogs is often lethal at the time of diagnosis. Therefore, there is a constant need for novel research on improvements of its characterization and treatment. Due to high cost and limited number of available dog patients, in vitro models of canine BC have been increasingly used for the last 25 years. In the present article, we present existing in vitro models of canine BC, including available simple (two-dimensional) and more complex (three-dimensional) models.

**Abstract:**

Bladder cancer (BC) constitutes approximately 2% of all spontaneously occurring cancers in dogs. It is characterized by a devastating clinical course in most cases, which emphasizes a constant need for the development of novel methods of disease characterization and treatment. Over the past years, advances in cell engineering have resulted in the development of various canine in vitro models of BC, emerging as complements for in vivo research. In this article, we aimed to review the available data on existing in vitro models of canine BC, focusing primarily on their characteristics, applications in veterinary medicine, as well as advantages and disadvantages. The most commonly used in vitro models of canine BC comprise immortalized cell lines grown as adherent monolayers. They provide an unlimited supply of research material, however, they do not faithfully reflect the conditions prevailing in vivo, since the spatial cellular interactions are lost. The importance of the three-dimensional (3D) features of solid tumors in relation to carcinogenesis or drug response process has resulted in the development of the first canine 3D models of BC available for in vitro research. So far, results obtained with in vitro and in vivo research should be interpreted together. With the constantly growing complexity of in vitro models of BC cancer, animal-based research might be reduced in the future.

## 1. Introduction

Bladder cancer (BC) constitutes approximately 2% of all spontaneously occurring cancers in dogs [1]. With estimates that 4–6 million pet dogs develop cancer in the United States annually, this equates to more than 60,000 cases of BC in dogs each year [2]. More than 95% of canine BCs are urothelial carcinomas (UCs), also known as transitional cell carcinomas (TCCs). Most canine BCs are characterized by adverse histopathological features at the time of diagnosis, such as muscle infiltration and high cellular grade [1,2,3]. Distant metastases are initially found in about 20% of newly diagnosed cases, leaving most dogs incurable [1,2,3]. The devastating clinical course and poor survival outcomes in considerable number of dogs with BC emphasize a constant need for development of novel research tools used for disease characterization and treatment, primarily resulting in survival outcomes improvements.

Though obvious benefits that could be gained from the research of canine BC in vivo, there are some inevitable limitations (e.g., high cost, long duration, and insufficient numbers of pet dogs to test even a fraction of the new drugs, especially when considering various possible drug combinations), making extensive in vivo research difficult [1,2]. Current strategies to overcome these challenges include increased utilization of in vitro models. Their resemblance to the primary tumors in the context of molecular behavior and genomic landscape has been increasingly evaluated in the past few years. Moreover, advances in cell engineering have resulted in the development of novel complex in vitro models of canine BC, emerging as unique complements for in vivo research.

In the present article, we aimed to review the available data on existing in vitro models of canine BC as a tool in veterinary research, focusing primarily on their characteristics, potential applications in veterinary medicine, and advantages and disadvantages. To the best of our knowledge, this is the first review summarizing the current evidence regarding this topic. 

## 2. Evidence Acquisition

We conducted a literature search using two electronic databases, namely, Pubmed and Scopus. The most recent search was performed on 28 January 2022. Screening of the literature was conducted using the following search string: (“bladder” OR “transitional” OR “urothelial”) AND (“cancer” OR “carcinoma” OR “neoplasm”) AND (“in vitro” OR “model” OR “cell*” OR “culture”) AND (“dog” OR “canine”). Auto-alerts in Medline were run, as well as reference lists of original articles and review articles for further eligible data. We exclusively included data regarding in vitro models of canine bladder TCC. Only papers in English were considered eligible without restrictions on publication year. The flow diagram of the study selection process is presented in Figure 1. 

## 3. Evidence Synthesis

### 3.1. Two Dimensional (2D) Models

The most commonly used two dimensional (2D) models of canine BC comprise immortalized cell lines and primary cell cultures grown as adherent monolayers in appropriate media. The first immortalized canine BC cell line, called K9TCC, was established by Knapp et al. in 1995 [4]. Since then, several novel canine BC cell lines have been developed and described in the literature [5,6,7,8,9,10]. Almost all of these cell lines were established from invasive and metastatic tumors, benefiting the investigation of late tumor progression and metastatic lesions. Their baseline characteristics were presented in Table 1.

The majority of canine BC cell lines express specific cancer-related markers, resembling those presented by the primary tumors in vivo. Due to differences in the development process (mainly related to primary tumor characteristics), the available canine TCC cell lines differ in terms of expressed biomarkers (Table 2). This allows for the selection of appropriate cell lines adapted to the research purposes, such as investigation of specific proteins and their potential impact on carcinogenesis or treatment response. Characterization of 8 canine BC cell lines (K9TCC, K9TCC-PU-AxA, K9TCC-PU-AxC, K9TCC-PU-Sh, K9TCC-PU-Mx, K9TCC-PU, K9TCC-PU-Nk, and K9TCC-PU-Pu) was provided by Dhawan et al. [5]. All cell lines revealed high expression of E-cadherin and cytokeratin. High cox-2 protein expression was present in all cell lines. The K9TCCAxA, K9TCC-PU-AxC, and K9TCC-PU-In cell lines were also characterized by high expression of p53 protein, whereas K9TCC, K9TCC-PU-Mx, K9TCC-PU-Nk, K9TCC-PU-Sh, and K9TCC-PU-Pu had low expression of p53 protein. Another available canine BC cell lines (K9TCC#1Lillie, K9TCC#2Dakota, K9TCC#4Molly, K9TCC#5Lilly) were characterized by Rathore et al. [7]. All cell lines highly or moderately expressed the cytokeratin. Cell proliferation marker Ki-67 was highly expressed in three of these cell lines, except K9TCC#4 Molly. Expression of kinase-tyrosine receptors (EGFR, PDGDR) differed between cell lines. PDGFR was more expressed in K9TCC#1Lillie, K9TCC#2Dakota, and K9TCC#4Molly than in the K9TCC#5Lilly. EGFR was moderately expressed in all tested K9TCC, whereas VEGFR seemed to be not expressed. Moreover, Cox-2 was highly expressed in all cell lines [7].

To date, only a few studies have provided data regarding high-throughput molecular characterization of existing canine BC cell lines. Initial molecular characterization (including genotypic data) of several canine BC cell lines was performed by the Flint Animal Cancer Center (FACC) [11]. Subsequently, Das et al. conducted whole exome sequence analyses on 33 canine cancer cell lines, including canine BC cell lines [12]. Authors provided a wide database of somatic mutations that can be explored for their role in the development and progression of canine BC [12]. Further investigations of the cellular biology through molecular characterizations of canine BC cell lines may provide valuable information regarding cancer biology and play a crucial role in predicting the variable treatment responses. Thus, in vitro analysis of drug sensitivity in a background of known protein coding somatic mutations could be used to correlate drug sensitivity to the observed genomic profile in further research.

In vitro studies using BC cell lines play a significant role in the novel drug discovery and development process, providing crucial data on drug effects in the early preclinical stages. Such information is of paramount importance in the decision-making process for drugs moving forward into more expensive and time-consuming in vivo clinical trials. Initial studies using canine TCC cell lines were extensively focused on non-selective cyclooxygenase inhibitors (Cox inhibitors, non-steroidal anti-inflammatory drugs—NSAIDs) and various chemotherapeutic agents [4]. Following the clinical trials with pet dogs, therapy with NSAIDs, with or without the addition of chemotherapeutics became the standard of care for canine invasive and metastatic TCC [13,14]. However, the overall median survival time for dogs that respond to NSAIDs and chemotherapy was still relatively short (up to a few months), which led to the search for novel therapeutic agents. In the past years, multiple studies including canine BC cell lines were conducted in order to evaluate the activity of novel anticancer agents (Table 3) [15,16,17,18,19,20,21,22,23,24,25]. Although many of them were not transferred to in vivo studies, novel therapeutic agents that could improve survival of dog pets with bladder TCC were also found. One of the most promising directions was molecular-targeted therapy using receptor tyrosine kinase inhibitors. As an example, Sakai et al. demonstrated that lapatinib (tyrosine kinase inhibitor of HER2 and EGFR) could inhibit canine BC cell growth in vitro [20]. Subsequently, Maeda et al. showed that compared to the dogs treated with piroxicam alone, those administered the lapatinib had a significantly greater reduction in the size of the primary bladder tumor and improved overall and progression-free survival [26].

Canine BC cell lines can be also used for investigation of other forms of anti-cancer therapy. As an example, Parfitt et al. investigated and characterized the radiosensitivity and capacity for cellular damage repair of canine BC cell lines [27]. Authors found that canine BC cell lines were moderately radioresistant and exhibited a high repair capacity. They concluded that larger radiation doses may be optimal for the treatment of naturally occurring BC in dogs [27]. In another study conducted by Maeda et al., significant differences in radiosensitivity between particular canine TCC cell lines (K9TCC and Bliley) were demonstrated. Bliley cell line was classified as radioresistant and K9TCC as radiosensitive, which might be used in further investigations on predicting individual response to radiation therapy in dogs with BC [28].

Conventional 2D cultures have several advantages supporting their important role in preclinical research. Research using cell lines is significantly less expensive than in vivo animal studies and provide an unlimited supply of material, which is widely available and easy to propagate under completely controlled and reproducible environmental conditions [29]. Nevertheless, 2D cell lines do not faithfully reflect the conditions prevailing in vivo since proper tissue structure and interactions with tumor microenvironment (TME), extracellular matrix (ECM), and host immune cells (ICs) are lost [30]. Moreover, the use of 2D cultures is usually restricted to one cell type, while tumors in vivo are frequently heterogenous in terms of the forming cell populations, being composed either by neoplastic cells or by stromal and ICs [29]. During each passage, cultured cells could experience genetic alterations due to selective pressure, which may lead to substantial changes in their phenotype. Unlimited access to oxygen and nutrients, unlike in vivo, can also induce accumulation of genetic changes that are not found in the primary lesions [29,30].

### 3.2. Three Dimensional (3D) Models

The importance of the three-dimensional (3D) features of solid tumors in relation to carcinogenesis or drug response process has prompted efforts to develop in vitro models mimicking in vivo tumor growth more precisely. Examples of 3D culture systems include multicellular aggregates grown as spheroids, scaffold-based models grown within polymer networks, and organoids defined as stem cell-containing self-organizing structures possessing multiple features of the original tumor [31,32]. Despite the increasing number of human studies using 3D models of BC, reports on canine models are still scarce.

The first 3D models of canine BC were established by Elbadawy et al. [33]. They generated four BC organoids using cells from urine samples collected from dogs with urothelial BC. Collected cells were mixed with natural polymer (Matrigel) and cultured with stem cell-stimulated medium. Established organoids had a spheroidal structure and a similar histology to naturally occurring BC in dogs. They were characterized by expression of urothelial cell markers and resembled the cellular architecture of invasive type of canine BC. Initial molecular characterization of established canine BC organoids has been performed and several novel genes were found to be specifically upregulated, being potential targets for novel therapies. Expression of several basal cell markers was found to be upregulated in generated organoids, suggesting that the cell origin of dog BC might be basal, which corresponds with poor response to chemotherapy in advanced stages. In a cell viability assay, the response to treatment with a range of anticancer drugs (e.g., cisplatin, vinblastine, gemcitabine, or piroxicam) was markedly different in each BC organoid, which forms the basis for further extensive research [33]. In addition, authors provided data on novel therapeutic agents, trametinib and verteporfin, which significantly inhibited the BC organoid viability. Additionally, trametinib induced basal to luminal differentiation of BC organoids, enhancing the sensitivity of cancer cells to carboplatin [34]. In another available study, the same research team demonstrated feasibility of performing 2D culture conditions using patient-derived 3D organoid cells without losing their characteristics, such as marker expression or stemness, creating a “2.5D” organoid canine BC model [35].

3D cell culture approaches hold great potential and offer complex systems for various purposes, such as disease modeling and investigation of anticancer drug efficacy. The similarities in the drug responsiveness among the 3D in vitro models and the in vivo models might largely be due to their similarities in enhanced cellular interactions via adhesion and secretion of soluble factors of tumors [31,36]. These new findings support the notion that cancer drugs which are currently being tested need to be screened using more complex tissue-like systems, rather than by using conventional 2D cultures that do not fully manifest features of in vivo tumors. However, 3D models are significantly more expensive than conventional 2D cultures, mainly due to the high cost of processing. In addition, current 3D models of BC are limited by a relatively narrow range of physical properties [31,36].

## 4. Conclusions

Although significant advances have been made over the past years, modeling the complexity of canine BC in in vitro models has not been completely successful. There are still various challenges, including the need for extensive molecular characterization of existing cell lines or creation of new reliable 3D models, incorporating multicellular cultures and diverse cellular microenvironments. So far, results obtained by in vitro and in vivo research should be interpreted together. With the constantly growing complexity of in vitro models of BC cancer, animal-based research might be reduced in the future

## Figures and Tables

**Figure 1 animals-12-00516-f001:**
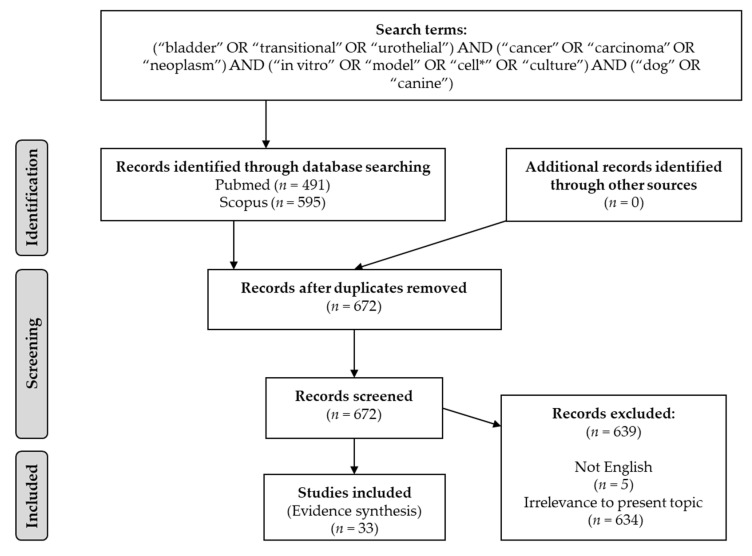
Flow diagram of study selection process.

**Table 1 animals-12-00516-t001:** Baseline characteristics of the available canine bladder cancer cell lines.

Cell Line Name	First Report Date	Development	Characteristics of Primary Tumor				Doubling Time	Reference (First Report)
			Breed of Origin	Age at Sampling	Gender	Pathological Data		
K9TCC	1995	Cultured cells from bladder tumor biopsy samples	Mixed breed	NR	Female	Invasive TCC	24 h	[4]
K9TCC-PU-AxA	2009	Cultured cells from bladder tumor biopsy samples	NR	NR	Female	Invasive TCC G3	23.5 h	[5]
K9TCC-PU-AxC	2009	Cultured cells from bladder tumor biopsy samples	NR	NR	Female	Invasive TCC G3	36.2 h	[5]
K9TCC-PU-In	2009	Cultured cells from bladder tumor biopsy samples	German Shepherd	NR	Female	Invasive TCC G3	41.2 h	[5]
K9TCC-PU-Mx	2009	Cultured cells from bladder tumor biopsy samples	German Shepherd	NR	Female	Invasive TCC G3	23.5 h	[5]
K9TCC-PU-Nk	2009	Cultured cells from bladder tumor biopsy samples	NR	NR	Female	Invasive TCC G3	58.4 h	[5]
K9TCC-PU-Pu	2009	Cultured cells from bladder tumor biopsy samples	NR	NR	Female	Invasive TCC G3	51.8 h	[5]
K9TCC-PU-Sh	2009	Cultured cells from bladder tumor biopsy samples	Collie	NR	Female	Invasive TCC G3	29.1 h	[5]
Bliley	2012	NR	Shetland Sheepdog	NR	Female	TCC	20 h	[6]
K9TCC#1Lille	2014	Cultured cells from bladder tumor biopsy samples	Pointer	16 years	Female	Invasive TCC	47.4 h	[7]
K9TCC#2Dakota	2014	Cultured cells from bladder tumor biopsy samples	Bichon Fries	13 years	Female	Invasive TCC	31.96 h	[7]
K9TCC#4Molly	2014	Cultured cells from bladder tumor biopsy samples	Maltese	10 years	Female	Invasive TCC	44.69 h	[7]
K9TCC#5Lilly	2014	Cultured cells from bladder tumor biopsy samples	Mixed breed	13 years	Female	Invasive TCC	48.3 h	[7]
LCTCC	2015	Cultured cells from bladder tumor biopsy samples	NR	NR	NR	TCC	NR	[8]
MCTCC	2015	Cultured cells from bladder tumor biopsy samples	NR	NR	NR	TCC	NR	[8]
MegTCC	2015	Cultured cells from bladder tumor biopsy samples	NR	NR	NR	TCC	NR	[8]
MonoTCC	2015	Cultured cells from bladder tumor biopsy samples	NR	NR	NR	TCC	NR	[8]
K9TCC-PU-An	2015	Cultured cells from bladder tumor biopsy samples	Scottish Terrier	NR	Female	Invasive TCC	NR	[9]
TihoDUrtTCC1506	2020	Cultured cells from bladder tumor biopsy samples	Labrador Retriever	10 years	Female	Invasive TCC	19.9 h	[10]

Abbreviations: G = grade; NR = not reported; TCC = transitional cell carcinoma.

**Table 2 animals-12-00516-t002:** Histological and molecular characterization of available canine bladder cancer cell lines.

Cell Line Name	Expression of Cancer-Related Markers									Available Molecular Data	Reference
	Uroplakin	Cytokeratin	E-Cadherin	Vimentin	Ki67	PDGFR	EGFR	COX-2	p53		
K9TCC	NR	High	High	Moderate	NR	NR	NR	High	Low	Array-based CGH, CNV analysis, transcriptome analysis	[4,5,9]
K9TCC-PU-AxA	NR	High	High	Moderate	NR	NR	NR	High	High	NR	[5]
K9TCC-PU-AxC	NR	High	High	High	NR	NR	NR	High	High	NR	[5]
K9TCC-PU-In	NR	High	High	Moderate	NR	NR	NR	High	High	Array-based CGH, CNV analysis	[5,9]
K9TCC-PU-Mx	NR	High	High	Low	NR	NR	NR	High	Low	Array-based CGH, CNV analysis	[5,9]
K9TCC-PU-Nk	NR	High	High	Moderate	NR	NR	NR	High	Low	NR	[5]
K9TCC-PU-Pu	NR	High	High	Moderate	NR	NR	NR	High	Low	NR	[5]
K9TCC-PU-Sh	NR	High	High	Moderate	NR	NR	NR	High	Low	Array-based CGH, CNV analysis	[5,9]
Bliley	NR	NR	NR	NR	NR	NR	NR	NR	NR	Deep exome analysis, transcriptome analysis	[6,11,12]
K9TCC#1Lilly	High	High	NR	Low	High	High	Moderate	High	NR	NR	[7]
K9TCC#2Dakota	High	High	NR	Low	High	High	Moderate	High	NR	NR	[7]
K9TCC#4Molly	Low	Moderate	NR	Low	Moderate	High	Moderate	High	NR	NR	[7]
K9TCC#5Lilly	Moderate	Moderate	NR	Low	High	Moderate	Moderate	High	NR	NR	[7]
TihoDUrtTCC1506	Low	High	High	Low	NR	NR	NR	High	Moderate	NR	[10]

Abbreviations: CGG = comparative genomic hybridization; CNV = copy number variations; COX = cyclooxygenase; EGFR = epidermal growth factor receptor; NR = not reported; PDGFR = platelet-derived growth factor receptor.

**Table 3 animals-12-00516-t003:** Examples of in vitro studies using canine bladder cancer cell lines to assess the efficacy of therapeutic agents.

Author	Therapeutic Agent	Cell Lines Used	Main Results	Reference
Knapp et al.	Piroxicam (COX-2 inhibitor)	K9TCC	Piroxicam had no direct cytotoxicity against canine BC cells Piroxicam increased cytotoxicity of chemotherapeutic agents	[4]
Galvao et al.	Gemcitabine + carboplatin (chemotherapeutic drugs)	K9TCC-PU: -AxA, -AxC, -Pu, -Sh	The combination of gemcitabine and carboplatin had synergistic antitumor effects on canine BC cells	[15]
Rathore et al.	AD198 (derivate of doxorubicin, chemotherapeutic drug)	K9TCC#Lillie, K9TCC#2Dakota, K9TCC#4Molly	AD198 inhibited cell viability of canine BC cells more efficiently as compared to doxorubicin at the same concentration	[16]
Gustafson et al.	Cyclopamine GANT6 (hedgehog signaling pathways inhibitors)	K9TCC, K9TCC-PU-Sh	Cyclopamine and GANT6 led to significantly decreased canine BC cells proliferation but had a smaller effect on apoptosis	[17]
Grayton et al.	KPT-185 KPT-335 (selective inhibitors of nuclear export)	Bliley	Canine BC cells were resistant to both drugs	[18]
Bourn et al.	Axitinib Masitinib (receptor tyrosine kinase inhibitors)	K9TCC#1Lillie, K9TCC#5Lilly	Axitinib and masitinib inhibited cell viability and increased apoptosis in a dose-dependent manner in tested canine BC cell lines	[19]
Sakai et al.	Lapatinib (tyrosine kinase inhibitor of HER2 and EGFR)	LCTCC, MCTCC	Lapatinib inhibited canine BC cell growth in a dose-dependent manner	[20]
Cronise et al.	Vemurafenib (BRAF inhibitor)	Bliley	BRAF mutant BC cell lines were insensitive to vemurafenib	[21]
Hurst et al.	Mavacoxib (selective COX-2 inhibitor)	K9TCC, K9TCC-PU: -AxA, - In, -Sh	Mavacoxib reduced cell viability in a dose-dependent manner in all tested canine BC cell lines	[22]
Byer et al.	Taurolidine (inhibitor of angiogenesis)	Bliley	Taurolidine showed significant effects on canine BC cell viability	[23]
Klose et al.	Metformin (biguanide antihyperglycemic agent)	TCC1506	Metformin inhibited the metabolic activity and cell proliferation of the canine BC cells	[24]
Korec et al.	Toceranib (multi-target receptor tyrosine kinase inhibitor)	K9TCC-PU-AxA, -AxC, -Nk, -Pu, -Sh	Toceranib at physiologically relevant concentrations has no direct anti-proliferative effect on canine BC cells	[25]

Abbreviations: BC = bladder cancer; COX = cyclooxygenase; EGFR = epidermal growth factor receptor; HER2 = human epidermal growth factor receptor 2.

## Data Availability

Not applicable.

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
