# Peer review of "Characteristics and Applications of Canine In Vitro Models of Bladder Cancer in Veterinary Medicine: An Up-to-Date Mini Review"

_animals, 2022, doi:10.3390/ani12040516_

Round 1
Reviewer 1 Report
General comment
This article is a review of an in vitro model of canine urothelial carcinomas. However, I felt that it was somewhat unclear what kind of information the authors wanted to provide to the readers. First of all, it is difficult to understand whether the target is human medicine or veterinary medicine. The tables are very concise and clear. However, in the text, the intent of the table is vague. What kind of audience (researchers) are you targeting? And what will the information in this manuscript be useful for? What advanced technologies are available to make up for the lack of in vitro models of canine urothelial carcinomas?
Specific comments
Introduction: It is recommended to clearly indicate whether the target of this study is an in vitro model of human bladder cancer, an in vitro model of canine bladder cancer, or both.
Line36: “Urothelial carcinomas" is not very familiar in veterinary medicine. In veterinary medicine, "transitional cell carcinoma (TCC)" is more familiar. As is the case with this meta-analysis, we recommend adding “also called transitional cell carcinoma" as an example so that veterinary researchers can read it. Does this meta-analysis include epithelial malignancies other than transitional cell carcinoma?
Table 1: Where is the explanation of the "G3" abbreviation? It is recommended that it be added to the bottom of the table.
Line77-89: The explanation in this paragraph is generally vague. If the target is human medicine, it is recommended to introduce the differences between the expression of markers in human bladder cancer. If the target is veterinary medicine, we recommend a brief explanation of the expression trends of each marker. Also, is it necessary to explain the reason for the difference in marker expression between cell lines?
Line93-108: In Table 3, therapeutic agents (especially molecular targeted drugs) are introduced. Line 93-108 is a paragraph related to Table 3, but it lacks content on therapeutic agents and focuses more on radiotherapy. I recommend that the description of therapeutic agents be more specific for balance.
Author Response
Dear Editor,
in reference to the decision of major revisions for the animals-1568665 manuscript, we are submitting a revised version of the article. All issues raised by Reviewers have been meticulously corrected. A detailed report on the amendments is presented below. If the Reviewers request additional language corrections, we will immediately send our manuscript to MDPI English Editing Service. Additionally, we decided to perform literature search once again as the last search was conducted almost 1.5 month ago.
#Reviewer 1
General comments
- First of all, it is difficult to understand whether the target is human medicine or veterinary medicine. The tables are very concise and clear. However, in the text, the intent of the table is vague. What kind of audience (researchers) are you targeting? And what will the information in this manuscript be useful for? What advanced technologies are available to make up for the lack of in vitro models of canine urothelial carcinomas?
Our response: We highly appreciate Reviewer’s feedback and valuable comments. The main aim of our article was to provide synthesized data regarding existing in vitro models of canine bladder cancer. We suppose that such data could help the researchers to select the particular model (specific cell lines, 3D model) for their research. Obviously, it is generally concise overview, however, all existing models were reviewed together with their characterization and examples of their utilization in veterinary research.
Specific comments
- It is recommended to clearly indicate whether the target of this study is an in vitro model of human bladder cancer, an in vitro model of canine bladder cancer, or both.
Our response: In the introduction section (Line: 55 - 58) we additionally stated that the target of this review are in vitro models of canine bladder cancer and their applications and examples of use in the veterinary research.
- Line36: “Urothelial carcinomas" is not very familiar in veterinary medicine. In veterinary medicine, "transitional cell carcinoma (TCC)" is more familiar. As is the case with this meta-analysis, we recommend adding “also called transitional cell carcinoma" as an example so that veterinary researchers can read it. Does this meta-analysis include epithelial malignancies other than transitional cell carcinoma?
Our response: We agree that “transitional cell carcinoma” is commonly used to describe the most common histological type of canine bladder cancer. However, some of the most recent studies prefer using “urothelial carcinoma”, as it is recommended in the human medicine. Bearing in mind that “TCC” is more familiar in veterinary medicine up to date, we added the proper statement and replaced “UC” with “TCC” in the main text (Line: 38 - 39). Only bladder TCCs were included in the present review, as it was additionally stated in the Evidence Acquisition section (line: 66 - 67)
- Where is the explanation of the "G3" abbreviation? It is recommended that it be added to the bottom of the table.
Our response: The explanation of this abbreviation was placed at the bottom of the Table 1.
- Line 77-89: The explanation in this paragraph is generally vague. If the target is human medicine, it is recommended to introduce the differences between the expression of markers in human bladder cancer. If the target is veterinary medicine, we recommend a brief explanation of the expression trends of each marker. Also, is it necessary to explain the reason for the difference in marker expression between cell lines?.
Our response: As it was mentioned before, the target is the veterinary medicine. Our primary aim was to demonstrate the expression of particular cancer-related markers in specific TCC cell lines, which might be helpful for the readers in the selection of TCC cell lines for their research (based on data provided in Table 2). We provided additional information corresponding to data in Table 2 in the main text (Line: 85 - 101)
- Line 93-108: In Table 3, therapeutic agents (especially molecular targeted drugs) are introduced. Line 93-108 is a paragraph related to Table 3, but it lacks content on therapeutic agents and focuses more on radiotherapy. I recommend that the description of therapeutic agents be more specific for balance.
Our response: Additional description of data presented in Table 3 was provided in the main text (Line 119 - 134)
Reviewer 2 Report
In the manuscript submitted by Lukasz Nowak et al., the authors summarize the status of canine bladder cancer research and describe the characteristics, advantages, and disadvantages of available preclinical models in 2D and 3D systems. The methods of literature search are well described in words and as a chart. The methods of literature searches are well documented in word and as a diagram, which gives a good overview but is not necessarily relevant for the validity of the review and can be included in the supplementary material.
The investigations of acquired cell lines in 2D models are listed properly alongside their molecular characterization. In addition, a collection of therapeutic agents tested in 2D models, and their in vitro efficacy is also displayed. However, it might be of interest to the reader to include additional outcomes from therapeutic agents tested in 2D models, especially if any were applied to canine bladder cancer or improved survival. If there is any data on individual drugs, please include them in table 3.
The overview of canine 3D models critically discusses the recent literature and concludes the limitations of the challenging model for BC cancer using canine 3D models.
For consistency of language, consider writing the entire text in BE or AE eg. tumor/tumour.
Overall, this paper summarizes the current literature in a clear and structured manner, and I endorse accepting this review after minor revisions.
Author Response
Dear Editor,
in reference to the decision of major revisions for the animals-1568665 manuscript, we are submitting a revised version of the article. All issues raised by Reviewers have been meticulously corrected. A detailed report on the amendments is presented below. If the Reviewers request additional language corrections, we will immediately send our manuscript to MDPI English Editing Service. Additionally, we decided to perform literature search once again as the last search was conducted almost 1.5 month ago.
#Reviewer 2
- The investigations of acquired cell lines in 2D models are listed properly alongside their molecular characterization. In addition, a collection of therapeutic agents tested in 2D models, and their in vitro efficacy is also displayed. However, it might be of interest to the reader to include additional outcomes from therapeutic agents tested in 2D models, especially if any were applied to canine bladder cancer or improved survival. If there is any data on individual drugs, please include them in table 3.
Our response: We highly appreciate Reviewer’s feedback and valuable comments. As recommended, we included additional data from in vivo studies using the therapeutic agents initially tested in 2D models. However, we wanted to avoid making Table 3 extensively large, so we included information in the main text (Line: 107 - 109, 112-119).
- For consistency of language, consider writing the entire text in BE or AE eg. tumor/tumour.
Our response: The entire text was checked and all BE words was replaced by AE words.
Round 2
Reviewer 1 Report
You responded appropriately to all my questions. No revision is required.